# High-Throughput Sequencing to Investigate Phytopathogenic Fungal Propagules Caught in Baited Insect Traps

**DOI:** 10.3390/jof5010015

**Published:** 2019-02-12

**Authors:** Émilie D. Tremblay, Troy Kimoto, Jean A. Bérubé, Guillaume J. Bilodeau

**Affiliations:** 1Canadian Food Inspection Agency, 3851 Fallowfield Road, Nepean, ON K2H 8P9, Canada; Emilie.Tremblay@canada.ca; 2Canadian Food Inspection Agency, 4321 Still Creek Dr, Burnaby, BC V5C 6S7, Canada; Troy.Kimoto@canada.ca; 3Natural Resources Canada, Laurentian Forestry Centre, 1055 Du P.E.P.S. Street, P.O. Box 10380 Québec, QC G1V 4C7, Canada; Jean.Berube@canada.ca

**Keywords:** insects, vectors, forest, fungi, metagenomics, HTS, oomycete

## Abstract

Studying the means of dispersal of plant pathogens is crucial to better understand the dynamic interactions involved in plant infections. On one hand, entomologists rely mostly on both traditional molecular methods and morphological characteristics, to identify pests. On the other hand, high-throughput sequencing (HTS) is becoming the go-to avenue for scientists studying phytopathogens. These organisms sometimes infect plants, together with insects. Considering the growing number of exotic insect introductions in Canada, forest pest-management efforts would benefit from the development of a high-throughput strategy to investigate the phytopathogenic fungal and oomycete species interacting with wood-boring insects. We recycled formerly discarded preservative fluids from the Canadian Food Inspection Agency annual survey using insect traps and analysed more than one hundred samples originating from across Canada. Using the Ion Torrent Personal Genome Machine (PGM) HTS technology and fusion primers, we performed metabarcoding to screen unwanted fungi and oomycetes species, including *Phytophthora* spp. Community profiling was conducted on the four different wood-boring, insect-attracting semiochemicals; although the preservative (contained ethanol) also attracted other insects. Phytopathogenic fungi (e.g., *Leptographium* spp. and *Meria laricis* in the pine sawyer semiochemical) and oomycetes (mainly *Peronospora* spp. and *Pythium* aff. *hypogynum* in the General Longhorn semiochemical), solely associated with one of the four types of semiochemicals, were detected. This project demonstrated that the insect traps’ semiochemical microbiome represents a new and powerful matrix for screening phytopathogens. Compared to traditional diagnostic techniques, the fluids allowed for a faster and higher throughput assessment of the biodiversity contained within. Additionally, minimal modifications to this approach would allow it to be used in other phytopathology fields.

## 1. Introduction

The Era of Globalization has dramatically and consistently increased international cargo shipments since 1970 [1,2]. Solid wood packaging material (SWPM), such as pallets, crates, and boxes are used to transport products all over the world. Bark and wood-boring insects, such as bark beetles, long-horned beetles, wood wasps, jewel beetles, weevils, and ambrosia beetles are often intercepted in SWPM [2,3,4,5,6]. Even with the implementation of International Standards for Phytosanitary Measures (e.g., ISPM No. 15), which states the need to treat wood products shipped abroad, in order to prevent the spread of insects and diseases, live wood-boring insects are still intercepted in SWPM, at the Canadian and American borders [7,8]. The emerald ash borer (*Agrilus planipennis*), brown spruce long-horned beetle (*Tetropium fuscum*), sirex wood wasp (*Sirex noctilio*), and the pine shoot beetle (*Tomicus piniperda*), are just a few examples of species recently introduced and established in Canadian forests [9,10,11,12,13].

The transmission of exotic phytopathogenic propagules, an important threat to forest health, figures among the many issues associated with the introduction of exotic insects in Canada, especially because some of these wood-boring insects proliferate within common North American tree species, such as pine and spruce [14,15,16,17]. Aside from killing or damaging trees, insects can also transmit different phytopathogenic species (e.g., fungal spores) to their respective plant host. A noteworthy example is the fungus *Ophiostoma ulmi*, the causal agent of Dutch elm disease, transported to elm trees by bark beetles [2,16]. Following its introduction by insect vectors (*Scolytus* spp.), this fungus devastated North American forests, more specifically the majority of elm trees, after it had also damaged elms in Asia and Europe [17,18]. Researchers have also previously reported potential links between insect and arthropod excreta and *Phaeoacremonium* spp., in grapevine infection cycles [19,20,21]. There also are numerous yeasts associated with insects. For instance, in addition to the fungus *Cladosporium* spp., the yeast *Candida* spp. have been linked to bark and rove beetles [22,23,24,25,26]. Oomycetes contain numerous plant pathogens responsible for considerable damage to the environment as well [17,27,28,29,30]. Their propagative structures can remain viable for further plant infection, even after ingestion and defecation by invertebrates [31]. For instance, chlamydospores of *Phytophthora ramorum* can still infect leaves after passing through the digestive tract of snails [31]. Oomycetes have also been associated with indirect interactions (positive or negative) with insects. For instance, while ants can transport *Phytophthora palmivora* and *Phytophthora megakarya* to cocoa trees, which can subsequently become infected [32], oviposition of the moth *Spodoptera littoralis* is enhanced after the *Phytophthora infestans* modifies the volatile compounds emitted by the host plant [33]. In contrast, the reproductive output by aphids is inversely proportional to the level of *Phytophthora* infection [34,35,36].

Insects can also benefit from mutualistic relationships with plant pathogens that overwhelm the plant’s defenses (e.g., thousand cankers disease), or induce a plant’s cell suicide response [15,37,38]. Contrarily, the damage caused by insects can indirectly predispose plants to microbial attacks. For example, in addition to the maize crop losses caused by the African pink stem borer (*Sesamia calamistis*) and the false codling moth (*Thaumatotibia leucotreta*), it was observed that the aflatoxin (produced by *Aspergillus* spp.) concentrations were proportional to the number of insects that came in contact with these crops, after storage [39,40,41]. Many bark and ambrosia beetles even rely on a fungal symbiosis to fulfill their nutritional needs [42].

While there are numerous studies of insects transmitting plant viruses [43,44,45] and bacteria [46,47], there is a need for additional research on associations between forest insects and microorganisms. Although there has been research on pinewood nematodes [38,48], ophiostomatoid fungi [15,16,32,49], and their insect vectors, there are likely more associations to be discovered, including transmission by vectors. For instance, additional fungal species never previously found to be associated with insect species might also be unexpectedly transmitted in this way.

The Canadian Food Inspection Agency (CFIA) conducts annual surveys using traps baited with semiochemicals, to detect non-native, wood-boring insects in high-risk areas, such as industrial and commercial zones [50,51,52,53]. Semiochemicals are communication chemicals to induce inter- or intraspecific interactions between organisms (e.g., decaying trees produce a kairomone that attracts bark beetles) [54]. They have been extensively studied for their ability to attract specific insects [48,55,56,57,58,59,60,61], especially for monitoring particular groups of beetles and the microorganisms associated with them. With the advances associated with high-throughput sequencing (HTS), scientists have used the power of metagenomics for the diagnosis of phytoviruses [43], fungi and oomycetes [62], and the detection of exotic fungi on asymptomatic live plant material that is imported into a given country [63]. In addition to its high sensitivity and capacity to sequence high volume of samples, HTS also allows for the analysis of hundreds of environmental samples, in a fraction of the time, compared to the traditional methods [64,65,66,67].

Taking advantage of a well-established, nationwide entomological survey, this project aimed to use a metagenomics approach to screen for the presence of phytopathogenic oomycetes and fungi contained in the preservative fluids originating from insect traps. The approach could potentially help forest pathology stakeholders to orientate surveys for disease monitoring and management, at a large scale, by filling in a gap in plant pathogen detection methods. In attempting to partially decipher the dynamics behind tree infection processes, valuable information from the insect trap fluid samples was extracted. The project screened for phytopathogens caught in baited insect traps as a primary evaluation of phytopathogenic fungal and oomycete propagules that were potentially transported—actively or incidentally—by wood-boring insects. Evaluation of commonalities between the province of collection, the year of collection, or the specific semiochemical used and the respective fungal or oomycete diversity, was also conducted.

## 2. Materials and Methods

### 2.1. Insect Traps

During the summers of 2013 to 2015, CFIA inspectors installed traps at 41 sites in industrial and commercial zones, landfills, and SWPM disposal facilities (Appendix A). These areas are the end-points for international SWPM and dunnage; they are considered to be high-risk areas for the introduction of non-native pests. At each site, 12-unit funnel traps, with wet collection cups (Synergy Semiochemicals Corporation, Burnaby, BC, Canada), were suspended between trees. Each trap’s collection cup was positioned at, approximately, 30 to 200 cm above ground, according to the height of the understory vegetation. Depending on the local temperatures, the baited traps were placed in forested areas, between March and April, and taken down at the end of September. The insects collected were part of a separate project and, therefore, were not used in this study. Rather, the preservative fluids—in which insects were kept until sample collection by inspectors—were recycled (fluids were formerly discarded) and analysed in this study. Resource limitation within the operational program prevented the usage and handling of the extra 12-unit traps, as a negative control (e.g., un-baited traps). To overcome this situation, assessment of the fungal communities (background noise) in the sampled areas was performed using Johnson and Barnes (JB) rainfall collectors (J. L. Johnson, personal communication). The samplers were used, as previously done by Barnes et al., Szabo et al. and Hambleton et al. [68,69,70,71]. More specifically, the JB spore collectors installed in the same location as the insect traps were used to dissociate the species collected by spore impaction (in both insect and spore traps), from the species caught through insect movement (only in insect traps). These JB samplers, employed as the negative controls, collected spores in suspension, in air and rainfall contents, at the surface of a membrane placed down the throat of the funnel-shaped traps. The JB spore collectors had a mesh screen to prevent insects from falling and clogging the funnel-shaped samplers. Collected fluids from the insect traps and the filter membranes from the spore traps were kept and analyzed, separately.

### 2.2. Semiochemicals

In 2013 and 2014, CFIA inspectors placed six traps per site. Half the traps were baited with one combination of lures, while the other half was baited with a different set. Inspectors attached one lure type to each trap, with each lure dispensed from individual release devices. Additional details pertaining to the semiochemicals (chemical composition, purity, packaging and release rate) used in this project have been provided in the Appendix A.

The first semiochemical combination (C_6_C_8_) contained ethanol, as well as aggregation pheromones of some long-horned beetles in the Cerambycinae subfamily [55]. The second semiochemical combination (UHR_E_AP) contained Ultra-High Release (UHR) ethanol and alpha-pinene, which are attractive to a wide range of bark and wood-boring insects [50,56,57]. Traps baited with C_6_C_8_ were suspended between coniferous or broadleaf trees, whereas traps containing UHR_E_AP were primarily placed between coniferous trees.

In 2015, CFIA inspectors implemented two new semiochemical sets, in order to target different insect taxa. The first semiochemical (i.e., the General Longhorn) was attractive to long-horned beetles in the Spondylidinae [58] and Lamiinae subfamilies [72], but could also capture various bark and ambrosia beetles, due to the addition of ethanol [59]. The second semiochemical set (i.e., the Pine Sawyer) was attractive to *Monochamus* (long-horned beetles) species from North America, Europe and Asia [48,60,61], but the inclusion of ethanol and alpha-pinene also made it attractive to bark and ambrosia beetles.

The semiochemical sets were deployed in different areas, depending on the forest type. In British Columbia, each site had four traps baited with the Pine Sawyer lures, and two traps baited with the General Longhorn. In Ontario and Quebec, 75% of the sites were in broadleaf forests or mixed forests, and all six traps were baited with the General Longhorn lure. The remaining sites, composed primarily of coniferous trees, were baited with the Pine Sawyer lure. In the Atlantic provinces (i.e., New Brunswick and Newfoundland and Labrador), each site had three traps with Pine Sawyer lure and three traps with the General Longhorn lure. Traps baited with the Pine Sawyer lure were suspended between two coniferous trees, whereas traps baited with the General Longhorn lure were placed between the coniferous or broadleaf trees. Lures were replaced approximately every 90 days.

The trapping fluid consisted of a mixture of 200 to 300 mL of United States Pharmacopeia (USP)/ Food Chemicals Codex (FCC) grade 1,2-propanediol (propylene glycol) (99.5% pure) (Fisher Scientific, Hampton, NH, USA), Denatonium benzoate or Bitrex^®^ = [Benzyl-diethyl (2,6-xylylcarbamoyl methyl) ammonium benzoate] (12.5 mg/L) (Sigma-Aldrich, Saint-Louis, MO, USA), and PhotoFlo 200 (0.5 mL/L) (surfactant) (Fisher Scientific) which was poured into the collection cups of each trap; the traps were spaced at least 25 m from each other [73,74].

### 2.3. Sample Processing and DNA Extraction

Samples (insect traps fluids and spore trap membranes) were collected every two to three weeks. The liquid contents (from insect traps and occasionally from the clogged spore traps) were poured onto fine-gauged sieves and the insects were removed because only the collection fluid was analysed in this study. The liquids were then filtered on 0.45-μm cellulose paper filters and stored at 4 °C, until processed. The filter papers (including those from spore traps) were cut in half, to preserve a section as a back-up, while the other half was put in Tris buffer, heated at 65 °C and sonicated (40 kHz). The solution containing the DNA was then centrifuged (10,000 rpm, 2 min) and extracted with the FastDNA kit, for soil (MP Biomedicals, Santa Ana, CA, USA). To remove the PCR inhibitors, purification of the extracted DNA was done using magnetic particles (Bio-Nobile, Östernäsvägen, Finland).

### 2.4. PCR, Purification and HTS

Then, PCR was performed, bidirectionally, to amplify DNA and add unique identifier (barcodes) to each sample, using Ion Torrent PGM fusion primers. The thermocycler used was a Veriti 96-well Fast (Fisher Scientific). Detailed sequences for fungi and oomycete fusion primers, as well as for the PCR cycling and the parameters, can be found in Tremblay et al. [62]. Fusion primers allowed for multiplexing the environmental samples and generating internal transcribed spacer 1 (ITS1) fungi and oomycete amplicons. In addition, when an oomycete band was visualized by electrophoresis, another PCR, targeting the adenosine triphosphate synthase subunit 9-nicotinamide adenine dinucleotide dehydrogenase subunit 9 (ATP9-NAD9) spacer, was performed, to allow for the proper downstream resolution of the *Phytophthora* species, considering that it is better-suited than the ITS, for resolving species within this genus [30,75,76,77]. All PCR included positive controls (i.e., fungal or oomycete or *Phytophthora* DNA isolates) and no-template controls (i.e., water). Products were visualized on a 1.5% agarose gel with a Gel Doc XR+ Gel Documentation System (Bio-Rad Laboratories, Inc., Hercules, CA, USA). Primer-dimers and other smaller-sized fragments (<100bp) were removed with Agencourt AMPure XP magnetic beads at a 0.7:1 beads:DNA ratio (Agencourt Bioscience, Beverly, MA, USA) [78]. Sequencing libraries were quantified with the Ion Universal Library Quantitation qPCR Kit (Life Technologies, Carlsbad, CA, USA) and the ViiA 7 Real-Time PCR System (Life Technologies), and then pooled at the equimolar concentration of 16 pM. The Ion Personal Genome Machine (PGM) Template OT2 Kit 400 bp (Life Technologies) and the Ion PGM sequencer (Life Technologies) were used to perform the HTS [79].

### 2.5. Bioinformatics

The raw data output from the sequencer was analysed using the pipeline, previously described by Tremblay et al. [62]. The FASTQ files were converted into sequence (FASTA) and quality score (QUAL) files, using fastqutils [80]. Quality trimming, based on the sequence quality and length, was done with Mothur (version 1.37.2) [81], using the trim.seqs function parameters minlength = 120, maxambig = 0 and maxhomop = 8. ITS extraction was done with ITSx (version 1.0.11) [82]. Using QIIME (version 1.7.0) [83], operational taxonomic unit (OTU) were picked using the open-reference method. OTU tables were generated using the QIIME, as well. To prevent missing the detection of rare species, OTUs with a single representative sequence were kept, although this was unlike the common practice of removing singletons [84,85]. Taxonomic assignment (method: BLAST) was done using the UNITE database (version 31.01.2016) [86] for fungi and the National Center for Biotechnology Information (NCBI) nucleotide database for oomycetes. The resolution power of the ATP9-NAD9 region for the *Phytophthora* species relied on a custom-built database, including the majority of all currently described *Phytophthora* species (NCBI accessions numbers JF771616.1 to JF772053.1 and JQ439009.1 to JQ439486.1) [30,87]. To evaluate species alpha diversity, evenness and the proportions of different organisms within the sample types, statistical analyses were done with R (version 3.1.3) [88], using the RAM package (version 1.2.1.3) [89]. Pairwise comparison between the diversity and the evenness associated with each semiochemical, was calculated through a *t* test, using the function “pairwise.t.test” from the R package Stats (Version 3.5.1) [88], and the *p* values were adjusted by the Benjamini & Hochberg [90] correction method. The RAM package was also used to generate sampling maps. Using the function “Adonis” from the R package Vegan (version 2.5-2) [91], permutational multivariate analysis of variance (PERMANOVA), using distance matrices (999 permutations, Jaccard method), was performed to evaluate the similarities between the Shannon True diversity of the subtracted fungi (see section below) and the area of collection (i.e., Eastern Canada, the West Coast or the Atlantic), the year of collection (i.e., 2013, 2014 or 2015), the lure employed (i.e., UHR_E_AP, C_6_C_8_, General Longhorn or Pine Sawyer), the lure set (i.e., UHR_E_AP and C_6_C_8_ versus General Longhorn and Pine Sawyer) and the province of collection (i.e., Quebec, Ontario, British Columbia, New Brunswick or Newfoundland and Labrador). Insufficient oomycete species remained after the subtraction, to perform an informative PERMANOVA analysis, so it was only performed on the fungi. To visualize the distribution of the type of trees that were used to hang traps at the collection sites, a data aggregation plot was built using the R package UpsetR version 1.3.3 [92].

### 2.6. Species Subtraction

Species subtraction, hereafter, refers to the dataset excluding species that were commonly detected in both the control and the insect traps. Insect trapping procedures were already established by the CFIA biologists and the sites were selected by CFIA inspectors, prior to this project [93]. Although CFIA entomologists screened and identified all wood-boring insects, due to time constraints, they only reported the non-indigenous insects captured during the survey. Considering the physical design of the insect traps, the passive collection of fungal spores suspended in air was inevitable. In an attempt to extract the unique features associated with insects attracted to the different types of semiochemicals, HTS data from the JB collector air samples—collected in a similar context as the insect traps (i.e., same areas, sampling time, sites, year, season, and so on)—was used to identify which fungal OTU were solely found in the insect traps. Although the JB collectors were passive spore samplers, through species subtraction, it was possible to determine which fungi or oomycetes species were unique to the insect traps and might be associated with an insect vector.

Following BLAST alignments of the OTU, with the respective reference databases, the remaining species were screened against a text-formatted database, to determine their biological fungal functions (available in Tedersoo et al. [94]), and the resulting file was parsed, to identify the fungi of interest, mostly those known to be plant pathogens and rot fungi. Ectomycorrhizal fungi, saprotrophs, mycoparasites, lichenized fungi, and most yeast were discarded.

To assess the depth of sequencing and diversity of the subtracted and negative control data (i.e., the spore trap), and to compare them with the original insect trap data, rarefaction curves were generated using the functions “diversity” and “rarefy” from the R package Vegan (version 2.5-2) [67,91]. This step was performed only on the fungal data as the oomycete dataset was too small to obtain relevant rarefaction curves.

## 3. Results

### 3.1. Trees Associated with the Samples Processed

A total of 108 samples originating from British Columbia, New Brunswick, Newfoundland and Labrador, Ontario and Quebec were collected, over three years (2013 to 2015). From those samples, 39 were baited with the UHR_E_AP, 36 with the C_6_C_8_, 17 with General Longhorn, and 16 with Pine Sawyer (Appendix A). Sample amplification results and efficiency by PCR can be found in Appendix A. As well, Appendix A presents a few types of trees surrounding the samples. However, the data does not fully represent the type of forest surrounding sampling sites, because the traps were only suspended from two trees.

### 3.2. Fungal Diversity Associated with the Four Respective Semiochemicals Following Species Subtraction

A total of 1,527 species remained once the species detected in the negative control were discarded, of which 368 species (approximately 25%) were common to all semiochemical types, 220 species (approximately 14%) were unique to the UHR_E_AP, 109 (7.1%) were unique to the General Longhorn, 99 (6.5%) were unique to the Pine Sawyer, and 116 (7.6%) were unique to the C_6_C_8_ lure (Figure 1a). Following the approach of Tedersoo et al. [94] to investigate fungal species functions, the data revealed the occurrence of some species of moderate concern, in each of the semiochemical treatments (Appendix A). In the Pine-Sawyer-baited samples, three rot fungi and six phytopathogenic fungi (including, *Ambrosiella ferruginea*, *Leptographium* sp., and the *Phaeoacremonium inflatipes*) were detected. From the General Longhorn samples, five rot fungi and eight phytopathogenic causal agents, including *Echinodontium tinctorium*, *Siroccocus conigenus*, and *Pucciniastrum circaeae* were detected. The C_6_C_8_ semiochemical treatment featured some unique fungi of interest as well—a single white rot, the yeast *Candida michaelii*, which is associated with the gut flora of the handsome fungus beetles (Endomychidae), and eight phytopathogens, including *Phyllosticta minima*, *Podosphaera clandestina*, and *Ciborinia Whetzelii*. Finally, the UHR_E_AP semiochemical had six rots, along with thirteen pathogenic fungi, namely the *Colletotrichum fructi*, *C. nymphaeae*, *Devriesia streliziicola*, *Erysiphe convolvuli*, *E. diffusa*, *Mollisia dextrinospora*, *Neoerysiphe galeopsidis*, *Podosphaera leucotricha*, *P. lini*, *Septoria gladioli*, *Stragonospora pseudopaludosa*, *Strelitziana mali* and the *Teratosphaeria xenocryptica*. The fungal sequence data obtained prior to species subtraction, including the rarefaction curves, can be found in Appendix A.

### 3.3. Oomycete Diversity Recovered from Three of the Four Semiochemicals after Species Subtraction

Following species subtraction, fifteen species remained, from which no unique oomycete species were detected in either the Pine-Sawyer-baited or the UHR_E_AP-baited traps (Figure 1b). Eleven species were unique to the General Longhorn, including the *Peronospora* sp., one was unique to the C_6_C_8_ (*Pythium sp.* CAL-2011e), and the three species were shared with the two latter semiochemicals (Appendix A). No *Phytophthora* sp. were unique to the insect traps, or any semiochemical treatment. The Oomycetes (ITS1) and *Phytophthora* spp. (ATP9-NAD9) sequence data obtained prior to species subtraction, can be found in Appendix A.

### 3.4. Comparison of Fungal Diversity by Semiochemical

The fungal species evenness (Shannon index [95]) (Table 1) was comparable between all lures but for the UHR_E_AP (w) (Appendix A; the letters w, x and y are used to show commonalities between the communities compared). The UHR_E_AP evenness was significantly different from the General Longhorn (*p* value = 0.017) and the Pine Sawyer (*p* value = 0.017) lures (x). The true diversity for fungi (Shannon index, per unit of number of species [95,96]) was evenly distributed among all semiochemical types, except for Pine Sawyer (y) (Appendix A). In addition, the Pine Sawyer diversity was significantly different from all lures but General Longhorn (x) (*p* value C_6_C_8_ = 0.006; *p* value UHR_E_AP = 0.002). Certain commonalities were also shared solely between the C_6_C_8_ and the General Longhorn lures (w).

### 3.5. Comparison of Oomycete Diversity by Semiochemical

Oomycete species evenness (Shannon index) revealed significant variations and common features between the different semiochemicals (Appendix A). Species evenness from the C_6_C_8_ shared features with the evenness of the UHR_E_AP (y) and the Pine Sawyer (w). The General Longhorn evenness was comparable only with the Pine Sawyer (x) (*p* value C_6_C_8_ = 0.0361; *p* value UHR_E_AP =0.0076). The UHR_E_AP and the Pine Sawyer evenness were also different (*p* value = 0.0293). The true diversity (Shannon) variation of the oomycetes is shown in Appendix A. There were similarities between the C_6_C_8_, Pine Sawyer and the UHR_E_AP lures (w). Other commonalities were unique between the General Longhorn and the Pine Sawyer diversity (x). Diversity index significantly varied between the General Longhorn and the C_6_C_8_ lures (*p* value = 0.046) and between the Pine Sawyer and the UHR_E_AP lures (*p* value = 0.046).

### 3.6. Variance of Fungal Diversity (PERMANOVA)

After discarding the fungal and oomycete species that were most likely passively or incidentally captured in the insect traps, we observed significant similarities between the Shannon true diversity and the different factors evaluated (Table 2). From the fungal data, 10% of the variation of the diversity was significantly explained by the province (*p* value = 0.001, F = 4.712 and df = 4), 5% (*p* value = 0.005, F = 3.105 and df = 3) by the lure type, 2% (*p* value = 0.014, F = 3.825 and df = 1) by the lure set and 2% (*p* value = 0.025, F = 3.412 and df = 2) by the year. The area of collection variation was not significant. Such observation was not possible for oomycetes because not enough species remained, following the species subtraction. Although, there were oomycete species unique to the Eastern zone, and aside from the *Saprolegnia* sp. SAP1 and the *Hyaloperonospora cochleariae*, all others were either *Peronospora* spp. (*P. farinosa, P. sparsa*, *Peronospora* sp. UPS F-119986, *P. viciae*, and *Peronospora* sp. isolate 079405,59), or *Pythium* species (*Pythium* aff. *hypogynum*, *Pythium* sp. CAL-2011f, *Pythium* sp. AvdB-2012, *Pythium* sp. P19300/1/3, and *Pythium* sp. BP2013k). One species—*Pythium* sp. BG02—was unique to the West Coast data but, there were no species uniquely associated with the Atlantic Region.

## 4. Discussion

This study used an innovative metabarcoding approach to investigate airborne and insect-vectored fungal and oomycete plant pathogen species, contained in preservative fluids, from baited insect traps. A unique aspect of this study was that it was the first time that the formerly discarded preservative fluids from multiple funnel traps of a given survey were analyzed. Additionally, the information extracted from the samples collected at a national scale was highly valuable and could identify many taxa, even at very low abundances (i.e., ≤10 OTU).

More specifically, the project aimed to expand knowledge on phytopathogenic fungi and oomycetes occurring in forests and surrounding areas, by studying insect trap fluid samples. Such an assessment was performed using the subtracted species datasets, in order to focus on microorganisms unique to the insect traps and, therefore, more likely to have been vectored by insects. To overcome the impossibility of using an un-baited insect trap, due to resources limitation, an original alternative negative control, which consisted of a spore trap placed at the same location as the insect traps, was used to assess and discard the fungal and oomycete species occurring in the background. Results, though ambiguous in certain cases, showed that there might be some novel insect–fungal relationships and effect of lure type that deserve further inquiry with a more specific set of semiochemicals. Traditional methods, such as cloning and culturing, make it difficult for scientists to screen species at a larger scale but, the analysis of insect fluids now provides the opportunity to perform a primary scan of unwanted species in a given environment, prior to performing any additional conventional testing for validation. Indeed, this project can provide a good overall assessment of the communities, without having to isolate all organisms for identification purposes. Given that, (i) identification with ITS1 is not always possible and (ii) confirmation with validated methods (e.g., qPCR) is essential for regulated species, this is a tool that regulatory agencies and other stakeholders could use for primary screening and disease monitoring, when combined with a proper un-baited insect trap and highly-specific lures. Areas identified would need to be further inquired, following critical detection, to ensure proper identification. This approach is essentially a general detection survey for non-indigenous pathogens that partially fills a large gap, which is a key step in the battle against invasive pathogens. By taking advantage of an established survey conducted by plant health inspectors, other than shipment of fluid samples to the laboratory, no additional resources were required, thereby, making this a cost-effective surveillance method. Just as air trap samplers are used to capture various pathogens in a given environment, our results suggested that insect traps can actively and passively gather worthy information from the environment.

Using ITS1 for fungal or oomycete communities profiling does not always allow for resolution below the genus level for certain organisms [97], and that is a direct consequence of the reliance of metagenomics on databases. Publicly available databases (e.g., NCBI nucleotide) are only partially curated and are incomplete, which reflects on the quality of the results generated. Curated databases, such as UNITE, contain much higher quality sequences, though that is to the cost of a much lesser diversity, due to the resources required to build and validate such a huge data load. Targeting a specific molecular marker (e.g., ATP9-NAD9) is also a limitation, when the sequences of only a few species are included in a given database. Additionally, certain genera contain very little divergence between their species, which represent a non-negligible risk for misidentification. For these reasons, the method described cannot be considered to be a diagnostic tool, but rather integrated into the toolbox, leading to a proper identification with additional standard techniques.

Still, like it was presented in the Appendix A, species subtraction filtered the data from background noise and highlighted numerous species with the potential to damage trees or other plants. Species detected in control traps were least likely to be vectored by wood-boring insects and were discarded, revealing that several potentially phytopathogenic entities still remained. Again, these clues suggest that the remaining species were more likely to be insect-transmitted, but lure-specificity and a negative control improvement would help confirm such hypotheses. Nevertheless, for instance, species unique to the various semiochemical types were from the genera composed of important plant pathogens. Among others, *Phoma glomerata*, the causal agent of blight, leaf spots, and fruit rot of many plants, was uniquely detected in the C_6_C_8_ samples, whereas, *Mycosphaerella areola* (mildew)—despite the fact that it typically infects wheat leaves [98,99], field peas [100], and cruciferous vegetables [101]—was only detected in the General-Longhorn-baited samples. This semiochemical is an aggregation pheromone for the long-horned beetles in the Spondylidinae [58] and Lamiinae subfamilies [72]. Similarly, the genera *Mortierella* and *Phyllosticta* (counting, *P. minima*: the causal agent of leaf spot in maple), which also contain numerous phytopathogenic species [102], were recovered from the C_6_C_8_ semiochemical, which is an aggregation pheromone for long-horned beetles in the Cerambycinae subfamily.

Data collected from the Pine Sawyer samples revealed fungi that are typically associated with insects such as the Ceratocystidaceae, *Ambrosiella* spp. and the *Leptographium* spp. [103,104,105]. Detection of *Leptographium,* a genus that includes the causal agent of blue stain in conifers, was not unexpected, given its known association with bark beetles [104,106]. For example, *L. piriforme* was vectored by *Tomicus piniperda* (exotic), as well as other native bark beetle species [107]. Interestingly, the proportion of the *Leptographium* species detected in the General Longhorn and the Pine Sawyer semiochemical traps, prior to species subtraction, were greater than that for the C_6_C_8_ and UHR_E_AP. This could suggest an association with particular insect groups or demes. As the phylogenetic analysis of *Leptographium* for species differentiation can be done using a combination of at least three genic regions (i.e., ITS2, β-tubulin and elongation factor-1α) [108], more sequencing data, or an alternate standardized assay (e.g., Sanger sequencing or specific molecular probes) would be required, to validate down to the species level (e.g., *L. piriforme*). Additionally, two considerable plant pathogen genera were observed solely within the Pine-Sawyer-lure-subtracted data: *Taphrina* (specifically *T. padi*, the causal agent of cherry fruit deformation) and *Phellinus* (*P. ferrugineovelutinus* is the causal agent of wood rot in alder and maple).

Given that some of the aforementioned fungal groups detected were less likely to actually encounter wood-boring insects, the presence of genera, such as *Phoma* spp., *Taphrina* spp., and *Mycosphaerella* spp., could also be explained by the occurrence of other insects caught into the insect traps. In fact, flies, wasps, bees, dragonflies, moths and other insects, which are regularly found by the CFIA inspectors in the traps (Troy Kimoto, *personal communication*), could have incidentally come in contact with infected plants, prior to be captured, and, therefore, could have transported fungal propagules and subsequently increased the biodiversity. For instance, honey bees are known vectors of the phytopathogenic bacteria, viruses, and fungi, during foraging activities [109,110,111,112,113]. Such observations directly illustrated the need to develop more specific lures.

Likewise, some insects attracted to the UHR_E_AP might be involved in the transmission of powdery mildews (*Erysiphe* spp.) and anthracnoses (*Colletotrichum* spp.), considering that these known pathogenic fungi were not detected in any other semiochemical treatment. However, because the addition of ethanol makes this lure attractive to a wide range of insects, there could be another reason for the detection of such pathogens in those samples, possibly, (i) the forest type, province of origin or weather conditions, given that mildews are wind and water dispersed [114], or (ii) the capture of other insects including pollinators, due to the broader attraction that is caused by ethanol. In fact, Rassati et al. [115] showed that the forest type and area of collection associated with an insect trap, influences the beetles’ communities, therefore, it is highly likely that those factors also influenced the microbial diversity at the same time.

Similarly, the General Longhorn semiochemical (which mainly attracts long-horned beetles and bark beetles) was associated with important phytopathogenic genera, including *Pucciniastrum* (among others, the rust *P. circaeae*), *Sirococcus* (specifically, *S. piceicola* and *S. conigenus*, two shoot blight causal agents), and *Trametes* (which reduces wood value as a decaying agent) [102]. Additionally, while some *Verticillium* species are pathogenic to insects [116], other phytopathogenic *Verticillium* species are transported by jewel beetles (e.g., *Agrilus* spp.) and bark beetles [117]. For example, in Europe, *V. dahliae* is transported to the *Quercus* spp. by the bark beetle *Scolytus intricatus* (which does not occur in North America) and the ambrosia beetle *Anisandrus dispar* (F.) [117]. Therefore, the presence of *V. isaacii* (which causes wilt in multiple plants) in samples baited with C_6_C_8_ might be due, in part, to ethanol contained in the lure, because it is highly attractive to the ambrosia beetles. For these samples, given that not all *Verticillium* species can be resolved using the ITS1 region [118,119], the detection of *V. isaacii* using ITS1 was not conclusive at this point. Still, alignments of ITS1 sequences were done with reference sequences and the possibilities included *V. isaacii*, *V. isaacii*, *V. tricorpus*, and *V. klebahnii*. Once again, for the above-mentioned reasons, complementary tests should be performed to validate these sensitive data.

There were similarities between the fungal (prior to species subtraction) communities originating from the traps baited with the UHR_E_AP and the C_6_C_8_ (Appendix A), and between the communities detected in the Pine Sawyer and the General Longhorn-baited samples. In contrast, differences occurred between the communities detected by the two former lure types, versus the two latter. Such variation could be due to the fact that there were fewer Pine Sawyer and General Longhorn (used for one season, in 2015) samples, compared with the number of UHR_E_AP and C_6_C_8_ (used for two seasons, in 2013–2014) samples, rendering comparison between the datasets unbalanced. In fact, the PERMANOVA analysis (post-subtraction) showed that 2% of the variance was explained by the year of collection and another 2% by the lure sets. In addition, another 5% could be explained by the lures, taken separately and 10% by the province of collection. The statistics demonstrated that understanding the dynamics behind insects and phytopathogens is actually highly complex and involves a number of components for consideration, at the same time. The forest type and the seasonal weather (temperature and rain) might also have influenced the communities retrieved.

Compared with fungi, the fewer number of identifiable oomycetes OTU, in this study, might be explained by the fact that the number of taxonomically described oomycetes is much lower [120,121,122]. One outstanding aspect of the oomycete analysis is that, following the species subtraction, most remaining species were unique to the General Longhorn samples (Appendix A) and none were recovered from the Pine Sawyer semiochemical. There were no *Phytophthora* species remaining after species subtraction but, *Pythium* (broad host-range, mainly affecting the roots or leaves) and the *Peronospora* species (broad host-range, mainly causing mildews) were dominant. Once again, following further inquiry and control and lure adjustments, this could contribute to the demonstration of novel observations between plants, insects and oomycetes.

The ITS1 *Phytophthora* OTU (i.e., non-subtracted data) revealed *Phytophthora foliorum* (affects *Rhododendron* spp. and causes Azalea leaf blight [123,124]), *Phytophthora* sp. *“kelmania”* (affects gerbera [125] and Christmas trees [126]), and *P. syringae* (has a wide host-range, causing numerous diseases [127]), but because this intergenic spacer did not contain sufficient variation for species resolution, unlike the ATP9-NAD9, these identifications were not conclusive. In contrast, the ATP9-NAD9 region allowed the identification of *P. cryptogea* (causes different diseases in numerous hosts [127]), *P. foliorum*, *Phytophthora* sp. *“kelmania”*, and *P. syringae* but, these were discarded, following species subtraction. This might indicate that the *Phytophthora*-insects associations are not as frequent, compared to the other oomycetes, such as *Pythium* spp.

Following species subtraction, there was a number of fungal species found that could degrade timber, some of which were associated with specific lures (Appendix A). Despite the fact that these fungi were not highly virulent, they could still damage or stain wood, thereby, reducing the timber marketability. The methodology used here appeared to have the capacity to detect more harmful organisms, if they had been present, because many genera containing virulent pathogens were detected.

Considering that the spore trap samples reached the most sequences per sample (see rarefaction curves in Appendix A), compared with both the insect traps and the subtracted insect trap datasets, it refuted the possibility of having mistakenly discarded some OTU, due to the spore trap under sampling. As the control samples were highly diverse, it was more likely that the species remaining after subtraction were actually unique to the insect traps. There were logically fewer species in the subtracted data versus the original one, which was likely why those remaining had a rarefaction curve approaching saturation. Interestingly, the fact that only a part of the original insect data was sequenced deeply enough (i.e., rarefaction curve saturation) suggested that, for future HTS runs, sequencing fewer multiplexed fusion primer samples, at a time, would probably yield a more representative diversity analysis. In contrast, it appeared that it was beyond sufficient for the spore traps, meaning more samples could be tested at once. In a resource-permitting context, sampling depth assessment with an un-baited insect trap should be performed.

Given that numerous yeast are commonly associated with insects [128], the presence of *Aureobasidium* sp., *Candida* sp., *Cystobasidium* sp., *Cryptococcus* sp., *Hannaella* sp., *Kluyveromyces* sp., *Rhodotorula* sp., *Torulaspora* sp., and *Wickerhamomyces* sp. in the unsubtracted dataset was expected (Appendix A). *Candida* spp. are natural biocontrols agents of fruit and vegetable pests [128,129,130], and are also associated with bark beetles [22,24,25,26]. Following species subtraction, our results showed the presence of *Candida michaelii* only in the samples baited with the C_6_C_8_ semiochemical, which primarily attracts long-horned beetles, but the addition of ethanol makes it attractive to bark beetles as well.

Furthermore, the fact that the entomopathogenic species *Colletotrichum nymphaeae* was retrieved from the samples is promising for entomologists, because they could consider using our method to screen for either insect pathogens, or new biological pest controls. As a matter of fact, beneficial fungal endophytes have been studied for their ability to help the plant’s defenses [131,132]. For example, *Beauveria bassiana* infection reduces populations of emerald ash borer (*Agrilus planipennis*) [133,134]. The gut-associated species detected (*Candida michaelii* [associated with the handsome fungus beetle]) demonstrated the robustness of our metagenomics approach in studying fungus–insect relationships, regardless of the niche they occupy. Additionally, this high-throughput method was more efficient compared with traditional assays, because many sites could be sampled simultaneously, due to the high volume of samples processed at a time, thus, increasing the likelihood of detecting new non-native fungal species.

Although some of the fungal genera or species that were detected in this project were already reported to be affiliated with insects, the results suggested that there may be other fungal and oomycetes species transported by insects. Based on the unique fungal and oomycete species detected within a given lure, our data suggested that there might be previously unrecorded associations between insects, fungi, and oomycetes that could be assessed by adding the analyses of the insects to the study. Each of the different semiochemicals employed in this study was attractive to a certain range of insect group but, inevitably, passive or incidental collection of other insects occurred (e.g., pollinators), which were contaminated by the phytopathogens that they came across—though they would otherwise never have transmitted it to the plant host—. The ethanol added also increased the chances for such events to happen. Therefore, to improve this biosurveillance tool, and like it was previously raised by Rassati, Faccoli, Petrucco Toffolo, Battisti and Marini [115], more research should be put into developing more specific semiochemicals. Still, there were auspicious hints towards the possibility that utilizing improved and more specific lures have the potential to provide valuable clues for insect–fungus associations. The use of a non-baited insect trap would also provide with more accurate background communities for future experiments.

## 5. Conclusions

All considered, making a direct link between the fungal species detected based on the semiochemical, the collection area, and insect vector is very complex when solely using the presented approach. Nevertheless, if an organism of potential interest was detected, this method would provide stakeholders with location data that would narrow the target area for requiring follow-up and proper identification surveys. These targeted surveys would involve searching for symptomatic hosts, collecting samples, and performing validated low-throughput assays, in order to overcome the metagenomics limitation due to databases. It is important to note that because the material studied consisted of airborne material, rather than symptomatic trees, the detection of a given species does not automatically translate to an occurrence of diseased trees.

Despite the fact that this method might lack in providing forest stakeholders with a definitive answer—due to the broader insect range attracted by ethanol and the alternate negative control used—the observations shed light on new potential insect–pathogen associations, worthy of more specific exploration. Yet, the approach to quickly screen species at a large geographic scale would be highly useful to (i) efficiently sample the environment for the presence of any new fungi or (ii) to obtain primary data, to provide guidance to those who monitor and manage phytopathogens over large jurisdictions (e.g., regulatory agencies). Finally, future research could examine the fungal communities associated with specific wood-boring insects, to determine if there are undiscovered relationships with these organisms and their host trees.

## Figures and Tables

**Figure 1 jof-05-00015-f001:**
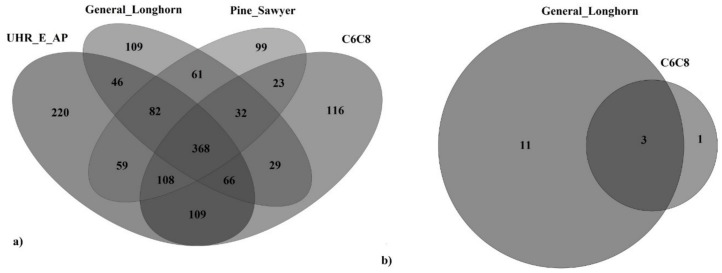
Venn diagram of (**a**) fungal species, shared or unique to the semiochemical type employed in the insect traps, after species subtraction and (**b**) oomycete species, shared or unique to the semiochemical type employed in insect traps, after species subtraction. All of the above were obtained by amplifying the ITS1 genic region.

**Table 1 jof-05-00015-t001:** *p* values of pairwise comparison between the diversity and evenness indices (Shannon) of fungal and oomycetes datasets, according to the lure employed. Significance calculated with *t* tests (Benjamini and Hochberg correction method). GL = General Longhorn, PS = Pine Sawyer, UHR = ultra-high release (UHR) ethanol and UHR alpha-pinene.

	Diversity		Evenness	
	Fungi	Oomycetes	Fungi	Oomycetes
GL vs C_6_C_8_	0.242	0.046 *	0.615	0.0361 *
PS vs C_6_C_8_	0.006 *	0.112	0.615	0.0924
UHR vs C_6_C_8_	0.586	0.504	0.065	0.8886
PS vs GL	0.104	0.504	0.932	0.5825
UHR vs GL	0.104	0.046 *	0.017 *	0.0076 *
PS vs UHR	0.002 *	0.125	0.017 *	0.0293 *

***** Significant p value calculated.

**Table 2 jof-05-00015-t002:** Permutational multivariate analysis of variance (PERMANOVA) of fungal diversity in relation to area of collection, or lure, or lure set, or province or year of collection, using distance matrices (999 permutations) and calculated according to the Jaccard similarity method. R^2^ = R-square statistic, F = F-statistic, df = degrees of freedom; *p* value calculated at a confidence interval of 95%.

Diversity	df	R^2^	F	*p* Value
Area ^a^	2	0.02	2.204	0.055
Lure ^b^	3	0.05	3.105	0.005 *
Lure Set ^c^	1	0.02	3.825	0.014 *
Province ^d^	4	0.10	4.712	0.001 *
Year ^e^	2	0.02	3.412	0.025 *

^a^ Sample collections occurred in Eastern Canada (i.e., Quebec and Ontario), the West Coast (i.e., British Columbia) or the Atlantic (i.e., New Brunswick and Newfoundland and Labrador); ^b^ Four different lures (semiochemicals) were used: The UHR_E_AP, C_6_C_8_, General Longhorn and the Pine Sawyer; ^c^ Corresponds to either the two lures used in 2013 and 2014 (i.e., UHR_E_AP and C_6_C_8_) or the two lures used in 2015 (i.e., General Longhorn and Pine Sawyer); ^d^ Samples were collected in the provinces of Quebec, Ontario, British Columbia, New Brunswick and Newfoundland and Labrador; ^e^ Years of collection were 2013, 2014 and 2015; * Significant *p* value calculated.

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
