# Peer review of "High-Throughput Sequencing to Investigate Phytopathogenic Fungal Propagules Caught in Baited Insect Traps"

_jof, 2019, doi:10.3390/jof5010015_

Round 1

Reviewer 1 Report

Line no. 18-25: Author may directly write the aim of study rather than giving a detail story.

Abstract must be in précised form it too much explanation of work done. Author may explain in materials and methods not in abstract. Bring it up to 250 words.

Materials and method written in very elaborative way. Eg. DNA Extraction method is so much elaborated.  Author must go through all the manuscript and do the needful corrections.

Discussion part also needs to be very much precise form.

Author Response

Line no. 18-25: Author may directly write the aim of study rather than giving a detail story. Abstract must be in précised form it too much explanation of work done. Author may explain in materials and methods not in abstract. Bring it up to 250 words.

Thank you. We improved the abstract and made sure it now is under 250 words. The final version is available below, and you can see the modification with tracked changes in the attached document.

“Abstract: Studying the means of dispersal of plant pathogens is crucial to better understand the dynamic interactions involved with plant infection. On one hand, entomologists rely mostly on both traditional molecular methods and morphological characteristics to identify pests. On the other hand, high-throughput sequencing (HTS) is becoming the go-to avenue for scientists studying phytopathogens. These organisms sometimes infect plants together with insects. Considering the growing number of exotic insect introductions in Canada, forest pest management efforts would benefit from the development of a high-throughput strategy to investigate the phytopathogenic fungal and oomycete species interacting with wood-boring insects. We recycled formerly discarded preservation fluids from the Canadian Food Inspection Agency annual survey insect traps and analysed more than 100 samples originating from across Canada. Using the Ion Torrent PGM HTS technology and fusion primers, we performed metabarcoding to screen unwanted fungi and oomycetes species including Phytophthora spp. Community profiling was conducted on the four different wood-boring insect attracting semiochemicals, although the preservative (ethanol) also attracted other insects. Phytopathogenic fungi (e.g., Leptographium spp. and Meria laricis) and oomycetes (mainly Peronospora spp. and Pythium spp.) solely associated with one of the four types of semiochemicals were detected. This project demonstrated that the insect traps’ semiochemical microbiome represents a new and powerful matrix for screening phytopathogens. Compared to traditional diagnostic techniques methods, the fluids allowed for a faster and higher throughput assessment of the biodiversity contained within. Additionally, minimal modifications to this approach would allow it to be used in other phytopathology fields.”

Materials and method written in very elaborative way. Eg. DNA Extraction method is so much elaborated.  Author must go through all the manuscript and do the needful corrections.

There are a lot of details that we added after reviewers from previous rounds requested them. For example, most reviewers asked for thorough details about the negative control spore traps. The same applies to our bioinformatics part where reviewers were requesting specific details to be put in the manuscript. We are conscious of the amount of information we provide in the Materials and Methods, and that is why we placed some of it in the Supplementary Materials already. However, given that it was almost a consensus between all reviewers to render this article more clear, we feel like it will be hard to take away some of those essential details.

Though, with regards to your specific comments on the DNA extraction section, another reviewer brought up a relevant comment which we applied and it may please you too: he/she mentioned that this DNA extraction section included more than just the extraction, which was right. So, we modified this header into “Sample processing and DNA extraction”, and added another header in the following paragraph, namely: “PCR, purification and HTS”. This much better represent what the material and method paragraphs are about.

Discussion part also needs to be very much precise form.

Again, much of the discussion includes paragraphs requested by reviewers (5 reviewers for round 1, 2 reviewers for round 2 and 2 reviewers for round 3). For instance, we were asked to discuss the limitations of using ITS and the pros and cons of relying on databases. We think this was essential to the paper and that is why we added it. Although, as you mentioned, it makes the discussion longer. Another example of a previous request is the addition of the PERMANOVA tables; it was a great addition to the article, but it did need to be further discussed and that also rendered the section longer. Another reviewer requested that we add a conclusion (it is actually optional for Journal of Fungi), which we also did because we think it better seals the work we are reporting here, but, again, it makes the section longer. The discussion is long essentially because there are i) a lot of results (prior and after the species-subtraction) and samples (more than 100, spread over three years); ii) the methodology is complex (semiochemicals, controls, validation, statistics, bioinformatics, multiple genic regions, multiple organisms screened for…) and has its limitations which must be stated; iii) the approach requires recommendations for further work based on what was learned here; and iv) results bring a whole new world of possibilities, which cannot be confirmed solely with this method, so emphasis must also be put on the fact that users need to use additional testing (e.g. qPCR, or other standard method) and fine tuning (for the control and the specificity of the semiochemicals for instance). We took time to go through the entire section and seek for parts that could be removed, but we could not find any in particular given that many paragraphs have been requested by reviewers from your journal in previous rounds. We hope our explanation will fulfill your request of a discussion in a very precise form.

Reviewer 2 Report

Few more minor comments:

- L169. This paragraph is not just about DNA extraction, please revise.

- If possible, move Table 2 and 3 to supplementary information.

I didn't like your answer about the mixed models (comment 3, 2nd round of revisions). I suggest you to fix this aspect, since you already performed the analyses.

Sincerely.

Author Response

Few more minor comments:

- L169. This paragraph is not just about DNA extraction, please revise.

You are absolutely right. We renamed the paragraph header “Sample processing and DNA extraction”, and we also added a “PCR, purification and HTS” header to the following paragraph. We believe it is now much better representing what those sections are about. Thank you.

- If possible, move Table 2 and 3 to supplementary information.

We initially put those tables in the Supplementary Materials but one of the reviewers (previous rounds of review) suggested two times that we move them into the main contents, so we did. The reviewer’s point of view made sense to us: he/she recommended putting everything that was obtained after species subtraction into the main part of this article, and to move the results obtained prior to species subtraction in the supplementary. We appreciate your input but we prefer to keep it the way it is now given that there is a logical structure behind it if we keep it like that.

I didn't like your answer about the mixed models (comment 3, 2nd round of revisions). I suggest you to fix this aspect, since you already performed the analyses.

Thank you. It appears that you may have misunderstood our reply which is likely because we were not clear enough the first time. The reviewer suggested performing a mix-model analysis. Given that it is a powerful test, we wanted to assess whether it was justified or not to do this statistical test on our data. The “pre”-assessment consisted of looking at variances between the different lures (inter-) and those associated with each of the lures (intra-) to evaluate if there was (or not) a difference in the variations (intra-) of each lure. If there had been, then the mix-model test would have been justified, however, we observed that there were some difference but that they were overall all comparable as they we all within the interval of confidence of each other. This is why we did not perform this mix-model test after all. Additionally, based on reviewers’ comments from previous rounds, we also performed PERMANOVA analyses.

Round 2

Reviewer 1 Report

Dear Author,

Thanks for providing the response to all suggested.

Author Response

Thank you very much for your positive comment.